# Graph Warp Module: an Auxiliary Module for Boosting the Power of Graph Neural Networks in Molecular Graph Analysis

## Abstract

Graph Neural Network (GNN) is a popular architecture for the analysis of chemical molecules, and it has numerous applications in material and medicinal science. Current lines of GNNs developed for molecular analysis, however, do not fit well on the training set, and their performance does not scale well with the complexity of the network. In this paper, we propose an auxiliary module to be attached to a GNN that can boost the representation power of the model without hindering the original GNN architecture. Our auxiliary module can improve the representation power and the generalization ability of a wide variety of GNNs, including those that are used commonly in biochemical applications.

## 1 Introduction

Recently, Graph Neural Network (GNN) is a popular choice of model in the analysis of molecular datasets in medicinal and material science. Many molecular datasets consist of molecular graphs with feature vectors associated to each atom, and numerous methods based on GNN has been proposed to date just for learning the features of chemical molecules (Wu et al., 2018; Duvenaud et al., 2015; Kearnes et al., 2016; Li et al., 2017; Gilmer et al., 2017; Shang et al., 2018), such as those pertaining to electrical conductivity and toxicity. One problem in the application of GNN to molecular datasets is the difficulty in reducing the training loss. Unlike in the applications of Deep Neural Networks (DNNs) to image datasets, the training loss of GNN on molecular dataset does not decrease consistently with the number of layers nor number of nodes per layers (cf. Fig. 4 in the appendix, thin dashed lines), and this seems to happen to numerous GNN architectures that are used in applications today (Duvenaud et al., 2015; Li et al., 2016; Kipf & Welling, 2017; Xu et al., 2019; Busbridge et al., 2018). Unfortunately, many strong techniques developed for deep CNNs such as ResNet (He et al., 2016) cannot be applied naively to GNN, because the tasks for GNNs are oftentimes fundamentally different in nature from that of standard DNN. For example, each graph data to be passed to the network can differ in size, and it is also often desired that GNN is equivariant (invariant under the reordering of vertices) in general. To the best of the authors' knowledge, there have not been many studies done to date that directly addressed the problem of training loss.

In this study, we propose graph warp module (GWM), a supernode (Li et al., 2017; Gilmer et al., 2017; Battaglia et al., 2018) based auxiliary module that can be attached to generic GNNs of various types to improve its representation power. The I/O of the auxiliary module is defined independently from the GNN to which it is attached, and the users can install the GWM just by adding a small segment of code.

Our GWM consists of three major components. The first component is *virtual supernode* (Li et al., 2017; Gilmer et al., 2017), which communicates with *all* nodes in the graph and promotes the remote message passing. The second and third components are attention unit (Vaswani et al., 2017; Veličković et al., 2018) and gating units (Cho et al., 2014). These adaptive weighting functions in the module help to adjust the flow of messages and deliver a message of appropriate strength to each node in the graph.

Our GWM can consistently improve the performance of various types of GNN on various types of dataset. In Fig. 1 we show the effect of GWM on the performance of four types of GNNs with the same embedding dimension and the same number of layers on four molecular graph datasets. As we

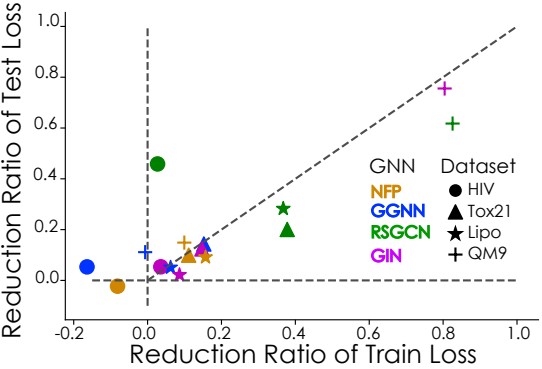

Figure 1: Train loss reduction and test loss reduction achieved by GWM on various model-dataset pair. The shape of each point presents the dataset used, and the color of each point presents the GNN model used. The horizontal axis denotes the ratio of reduced training loss, and the vertical axis denotes that of the test loss. That the x-coordinate of a point is positive implies that the attachment of GWM improved the train performance for the corresponding model-dataset pair. That the y-coordinate of a point is positive implies that GWM improves the test-performance.

can see in the figure, the attachment of GWM reduces both train loss and test loss for all but three model-dataset pairs. The GWM provides not only more representation powers (less train loss), but also better generalization performances (less test loss) for various GNNs.

As we will show in section 4.4, we can further improve the positive effect of GWM using hyperparameter optimization softwares such as Optuna (Akiba et al., 2019).

Our contributions are as follows:

1. We introduce GWM, an auxiliary module that can help improve the representation power of the GNNs that are designed for the analysis of small graphs.

2. We show that the attachment of GWM can improve both the representation power and the generalization ability of various GNN models on many popular molecular graph datasets.

## 2 RELATED WORK

### 2.1 VIRTUAL SUPERNODE

A common challenge in the application of GNNs to a graphical dataset is the difficulty in propagating the information across remote parts of graphs. Previously proposed solutions include sub-sampling (Hamilton et al., 2017) and pooling of neighbor nodes (Ying et al., 2018). However, these clustering approaches are not too effective on the analysis of small graphs, in which every node can have a strong influence on the graph label.

In this study, we use supernode (Gilmer et al., 2017; Li et al., 2017; Pham et al., 2017) to promote the global propagation of the information in molecular graphs. By adding a supernode to a graph, we can allow any pair of nodes in the graph to communicate through the supernode in one hop. Battaglia et al. (Battaglia et al., 2018) discusses a framework of GNNs that generalizes the supernode-augmented GNNs. One advantage of the supernode-based approach is that we can modify the network architecture while keeping the original GNN model intact. However, naive addition of a supernode to a graph can potentially lead to inadvertent over-smoothing of information propagation (c.f. (Li et al., 2018)). In our study, we therefore make the supernode a *module* by combining it with a *gated message passing mechanism*. This auxiliary module enables us to regulate the amount and type of information that is propagated through the feature space of the supernode.

### 2.2 MESSAGE PASSING AND ATTENTION/GATE MECHANISM IN GNN

The supernode in our GWM transmits information using the mechanism of message passing neural network (MPNN) (Gilmer et al., 2017), which is defined recursively as follows by composing multiple layers of the form:

$$h_{\ell,i} = \mathscr{F}_\ell \left( \{ h_{\ell-1,j} ; j \in N(i) \cup \{i\} \} \right) \tag{1}$$

where $i, j$ are indices of nodes in a graph. $h_{\ell,i}$ is the feature vector of the node $i$ at the $\ell$th layer, $N(i)$ is the neighborhood of the node $i$, and $\mathscr{F}_\ell$ is an appropriate choice of function that updates the

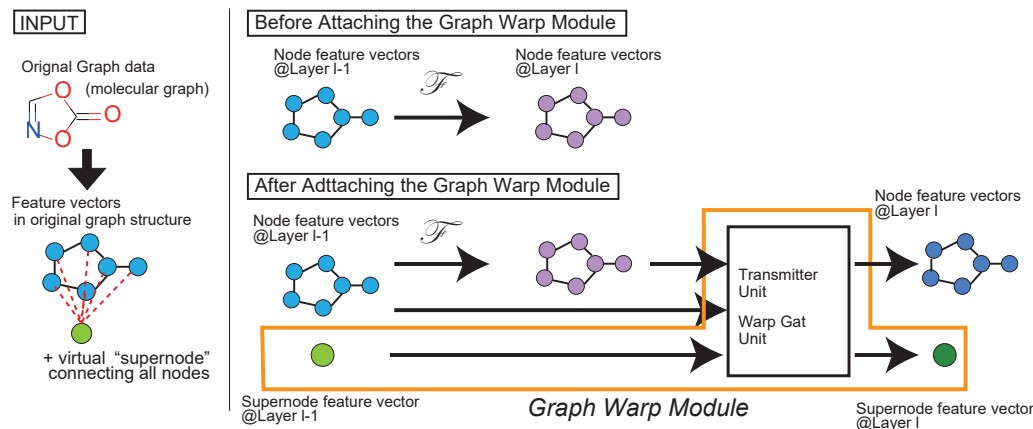

Figure 2: The overview of the proposed Graph Warp Module (GWM). A GWM consists of a supernode, a transmitter unit, and a warp gate unit. A GWM can be added to the original GNN as an auxiliary module. At each layer, the supernode and the main network communicate through the transmitter and the warp gate.

feature vectors of the previous layer. That is, MPNNs work by passing the information of each node to its neighbors in a recursive manner. Various methods are proposed for the choice of $\mathscr{F}$ and for the method of pooling the information of the neighbors of each node (Schlichtkrull et al., 2017; Kipf & Welling, 2017; Li et al., 2016; Bruna & Szlam, 2014; Duvenaud et al., 2015).

Attention mechanism is a mechanism that helps the network regulate the importance of each node/edge in message passing (cf. (Wang et al., 2018)). A Relational GCN (Schlichtkrull et al., 2017) assumes that the aggregation weights of $h_{\ell-1,j}$ is fixed a priori in all $\mathcal{F}_\ell$. With such architectures, however, one cannot regulate the higher order correlation amongst the outputs from each node. Graph Attention Networks (GATs) (Veličković et al., 2018) introduce a self-attention mechanism (Vaswani et al., 2017), which is equipped with a trainable set of weights that controls the importance of edges for each node. The relational graph attention network (RGAT) (Busbridge et al., 2018) also builds upon GAT and constructed multiple types of attentions derived from relation-type-wise intermediate node representations. Finally, a GRU (Cho et al., 2014) is a gating mechanism originally introduced for recurrent neural networks. Gated Graph sequence Neural Networks (GGNN) (Li et al., 2016) are the first to apply GRUs to the GNNs, and their method aims to introduce a recurrence relation between successive layers. Although not in the form of super-node, a very recent paper (anonymous, 2019) uses a mechanism that combines GGNN and self-attention to capture both local and global relationships of a graph structure of program source codes. Our GWM is equipped with both multi-relational attention mechanisms and GRUs to grant the module greater flexibility for the transmission of message between supernode and the bulk nodes.

## 3 GRAPH WARP MODULE

Our Graph Warp Module (GWM) is made of three building blocks: (1) a supernode, (2) a transmitter unit, and (3) a warp gate unit (Fig. 2). In a GWM-attached GNN, information is propagated across the graph through *communication* between the supernode and the original (main) GNN at each layer. Messages from the supernode and the main GNN are transmitted to the warp gate through the transmitter unit, and the results of the communication are passed back to the module/main network through the warp gate units. In this section, we describe the Graph Warp Module in detail, and present the motivation of our design.

### 3.1 PREMISE: VANILLA GNN AND ITS I/O

Before describing our GWM, we need to present the I/O notation for the family of GNNs we consider, and explain how they will be used when a GWM is attached to a GNN. We denote an arbitrary

graph with the edge set $E$ and the node set $V$ as $G = (V, E)$. We will label the nodes in $V$ as $i = 1, 2, ...|V|$, and represent each edge as a pair of nodes in $V$. The adjacency matrix $\mathcal{A} \in \mathbb{R}^{|V| \times |V|}$ is a matrix whose $(i, j)$th entry is the weight assigned to the edge between the node $i$ and the node $j$. Each instance of *data* passed to the GNN is a set of input feature vectors. We denote an input feature vector associated with node $i$ as $x_i$. The type of GNN that we consider computes the output recursively by applying a composition of smooth functions $\mathscr{F}_\ell$ to $x_i$s. With the understanding that $x_j = h_{0,j}$, let $h_{\ell,i} = \mathscr{F}_{\ell-1,i}(h_{\ell-1,j}; j \in V) \in \mathbb{R}^d$ be the vector of features to be assigned to the $i$th node by the $\ell$th layer of the GNN. When the GNN is operating on its own without the attachment of a GWM, the GNN updates a feature vector using $h_{\ell,i} = \mathscr{F}_{\ell-1,i}(h_{\ell-1,j}; j \in V) \in \mathbb{R}^d$. Finally, the GNN reports some form of the aggregation of $\{h_{L,i}; i \in V\}$ as the final Readout output.

When a GWM is attached to the GNN, the main(bulk) GNN is requested to report $\mathscr{F}_{\ell-1,i}(h_{\ell-1,j}; j \in V) \in \mathbb{R}^d$ as the message from the $\ell - 1$th main layer to the module, where it is treated as an element in the *intermodule hyperspace* and is mixed with the transmission from the supernode. The GWM will return the mixed message $h_\ell$ back to the $\ell$th layer of the main GNN. At the same time, the GWM requests a transmission message from the main GNN to the $\ell$th supernode. The module will mix the transmission and the message from the $\ell - 1$th supernode and return the mixed message $g_\ell$ to the $\ell$th supernode. The final output is produced by aggregating $\{h_{L,i}; i \in V\}$ and $g_L$.

### 3.2 SUPERNODE

A supernode is a special node that is connected to nodes in the original graph to promote global information propagation across the network (Fig. 2). A supernode is to be prepared for each $\ell$th layer of the main GNN, and we associate a feature vector $g_\ell$ to the supernode at the $\ell$th layer. At each layer, the transmitter requests the following from the supernode: (1) a message $\mathscr{G}_\ell(g_\ell)$ for the $\ell + 1$th layer and (2) a transmission to the main network, where $\mathscr{G}_\ell$ is an appropriate choice of a smooth function.

Because a supernode is a superficial variable, we must initialize $g_0$ manually. For instance, we can use some form of aggregation of the global graph features (e.g. a number of nodes or edges, graph diameter, girth, cycle number, min, max, histogram, or an average of input node features, . . . ). A detailed example of the aggregated feature is presented in the appendix.

### 3.3 TRANSMITTER UNIT

The transmitter unit handles the communications between the main GNN module and the GWM (Fig. 3). The transmitter module is responsible for translating the messages from the recipient into a form that can be mixed in the intermodule hyperspace. We will use multiple types of messages and thus use a separate attention mechanism for each type of message. Before transmitting messages from the main GNN to the supernode, the transmitter uses the $K$-head attention mechanism to aggregate messages of each type. We enumerate the components included therein:

- $m_{\ell,k}^{\text{main}\to\text{super}}$: aggregated message of head $k$ from the main GNN to the supernode at layer $\ell$.
- $h_\ell^{\text{main}\to\text{super}}$: transmission from the main GNN to the supernode, derived from $m_{\ell,k}^{\text{main}\to\text{super}}$.
- $g_\ell^{\text{super}\to\text{main}}$: transmission from the supernode to the main at layer $\ell$.

The transmissions are to be constructed from the following set of equations. For a vector $v$, we use $v_{m:n} \in \mathbb{R}^{(m-n)d}$ to denote the concatenation of the vectors $v_m, v_{m+1}, ... \in \mathbb{R}^d$. Throughout, we use capital letters to denote the trainable coefficients.

$$h_\ell^{\text{main}\to\text{super}} = \tanh\left(W_\ell\, m_{\ell,1:k}^{\text{main}\to\text{super}}\right) \in \mathbb{R}^{D'}, \tag{2}$$

$$m_{\ell,k}^{\text{main}\to\text{super}} = \sum_i \alpha_{\ell,i,k} U_{\ell,k} h_{\ell-1,i} \in \mathbb{R}^{D'}, \tag{3}$$

$$\alpha_{\ell,i,k} = \text{softmax}\left(h_{\ell-1,i}^T A_{\ell,k} g_{\ell-1}\right) \in (0, 1), \tag{4}$$

where $\alpha_{\ell,i,k}$ denotes an attention weight of the $i$th node at the $k$th head (type) and the $l$th layer.

The transmission from the supernode to the main is simply given by:

$$g_\ell^{\text{super}\to\text{main}} = \tanh\left(F_\ell g_{\ell-1}\right) \in \mathbb{R}^D. \tag{5}$$

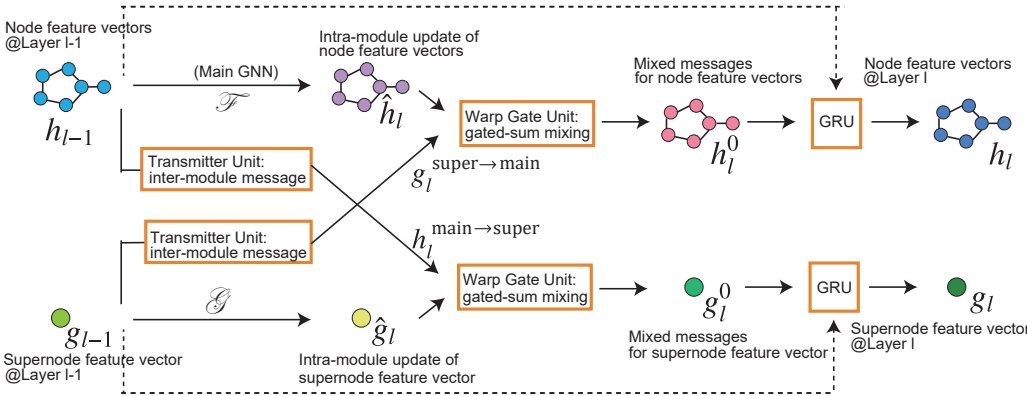

Figure 3: Details of the GWM computations.

There is no analogue of $m$ for the supernode because we are not considering a set of messages of different types to be transmitted from the supernode.

### 3.4 WARP GATE

The warp gate is responsible for merging the transmitted messages and for passing the results to the supernode and the main network through self recurrent units. The gate uses *warp gate coefficients* to control the mixing-rate of the messages. The components of the warp Gate are:

- $h_\ell^0$: inputs to the GRU unit at $\ell$th layer that transmits the message to the main GNN.
- $g_\ell^0$: inputs to the GRU unit at $\ell$th layer that transmits the message to the supernode.
- $\hat{h}_{\ell,i}$: the message $\mathscr{F}_{\ell-1,i}(h_{\ell-1,k}; k \in V) \in \mathbb{R}^D$ from the $\ell - 1$th layer of the main network, expressed in the intermodule hyperspace.
- $\hat{g}_\ell$: the message $\mathscr{G}_{\ell-1}(g_{\ell-1}) \in \mathbb{R}^{D'}$ from the $\ell - 1$th supernode, where $\mathscr{G}$ is an appropriate smooth function with outputs in the intermodule hyperspace.
- $z_{\ell,i}$: tensor of warp gate coefficients for the transmission from the supernode to the main GNN.
- $z_{\ell,i}^{(S)}$: tensor of warp gate coefficients for the transmission from the main GNN to the supernode.

The module then mixes the transmissions and the messages from the previous layer by applying the following gated interpolations:

$$h_{\ell,i}^0 = (1 - z_{\ell,i}) \odot \hat{h}_{\ell-1,i} + z_{l,i} \odot g_\ell^{\text{super}\to\text{main}} \in \mathbb{R}^D \,, \tag{6}$$

$$g_\ell^0 = z_\ell^{(S)} \odot h_\ell^{\text{main}\to\text{super}} + (1 - z_\ell^{(S)}) \odot \hat{g}_\ell \in \mathbb{R}^{D'} \,, \tag{7}$$

$$z_{\ell,i} = \sigma\left(H_\ell \tilde{h}_{\ell,i} + G_\ell g_\ell^{\text{super}\to\text{main}}\right) \,, \quad z_\ell^{(S)} = \sigma\left(H_\ell^{(S)} h_\ell^{\text{main}\to\text{super}} + G_\ell^{(S)} \hat{g}_\ell\right) \,, \tag{8}$$

where $\sigma$ is a nonlinear function whose range lies in $[0, 1]$. Finally, the warp gate returns the mixed messages to the main GNN and the supernode through gated recurrent unit (GRU) s:

$$h_{\ell,i} = \mathbf{GRU}\left(h_{\ell-1,i}, h_{\ell,i}^0\right) \in \mathbb{R}^D \,, \quad g_\ell = \mathbf{GRU}\left(g_{\ell-1}, g_\ell^0\right) \in \mathbb{R}^{D'} \,. \tag{9}$$

As for the structure of GRU, we used the original design introduced by (Cho et al., 2014).

As we will show with ablation studies (Sec. 4.4), every component of GWM is essential in making the module work. The Attention coefficients (Eqs.(3,4)) are important because the amount and the type of information that must be transmitted to remote nodes may differ for different nodes. The Gating coefficients (Eqs.(6-8)) are important because we want to regulate the transmission from each

node in the graph to the supernode and vice versa. We use different recurring units (Eq.9) for the transmission from the module to the supernode and the transmission from the module to the main network for each layer because the amount of the information that must be reinforced may differ for the main network and the supernode at each layer.

### 3.4.1 COMPUTATIONAL COMPLEXITY

Let $V$ be the vertex set, $K$ be the number of attention heads, and $D$ be the dimension of the node embedding. Then the additional computational cost incurred by the attachment of GWM is at most $O(|V|KD^2)$. As for the actual computation time, GWM attached module consumes approximately double the time of the original unattached version.

## 4 EXPERIMENTS

In this section, we present our experimental results on multiple molecular graph datasets, testing the efficacy of the GWM for graph regression tasks and graph classification tasks.

### 4.1 DATASETS

We used four datasets collected in MoleculeNet (Wu et al., 2018). These datasets are described in the SMILES string format, which admits the graph representations we described above. For details, please see (Wu et al., 2018).

For the graph regression tasks, we used the QM9 dataset and the Lipophilicity (LIPO) dataset. QM9 is a dataset with numerical labels, containing about 133K drug-like molecules with 12 important chemical-energetic, electronic, and thermodynamic properties, such as HOMO, LUMO, and electron gaps. The LIPO dataset is another numeric-valued dataset, containing the solubility values of roughly 4K drug molecules. Each instance of data in these datasets is a pair of a molecular graph and a numerical value(s): the 12 chemical properties in the QM9 dataset, and the solubility in the LIPO dataset. For both datasets, the task is to predict the numerical value(s) from the molecular graph. We evaluated the performance of the models using mean absolute errors (MAEs). We report the averaged MAE over 12 sub-tasks (properties) for QM9.

For the graph classification tasks, we used the Tox21 and the HIV datasets. The Tox21 dataset contains about 8K pairs of molecular graph and 12 dimensional binary vector that represent the experimental outcomes of toxicity measurements on 12 different targets. The HIV dataset contains roughly 42K pairs of molecular graph and binary label that represent the medicinal effect of the molecule. For these datasets, the task is to predict the binary label(s) from the molecular graph. For these tasks, we use ROC-AUC values as a measure of performance. We report the averaged ROC-AUC over 12 sub-tasks (targets) for Tox21.

Throughout, we used the train/validation/test data splits of the "scaffold" type, which is considered by (Ruddigkeit et al., 2012; Ramakrishnan et al., 2014) as the difficult type for test predictions. Please find the appendix for details.

### 4.2 CHOICES OF THE MAIN GNN MODELS AND IMPLEMENTATIONS

We test GWMs on various GNN models. Neural Fingerprints (NFP) (Duvenaud et al., 2015) and Weavenet (Kearnes et al., 2016) are relatively classical baselines. A Gated Graph Neural Network (GGNN) (Li et al., 2016) is a strong GRU-based GNN. Renormalized Spectral Graph Convolutional Network (RSGCN[1]) (Kipf & Welling, 2017), a popular GNN model approximating a CNN for graphs (Defferrard et al., 2016). The relational graph attention network (RGAT) (Busbridge et al., 2018) uses multiple attention mechanisms for a set of edge types. Graph Isomorphism Network (GIN) (Xu et al., 2019) employs multi layer perceptrons within each layer for richer transformations.

We implement all models in Chainer (Tokui et al., 2015). In the readout layer, we first aggregate all information from the main nodes in the same way as in the original paper, concatenated the result with the features from the supernode, and passed the concatenated tensor to a fully connected layer.

---

[1]Referred as RSGCN in Chainer-Chemistry package, but often simply referred as "GCN" in several papers.

| Model Name (emulating) | Attention | Gatings | GRUs |
|---|---|---|---|
| Simple supernode (Li et al., 2017; Pham et al., 2017) | — | — | — |
| NoGate GWM (Gilmer et al., 2017) | ✓ | — | ✓ (supernode only) |
| Proposed GWM | ✓ | ✓ | ✓ |

Table 1: Supernode-based models validated in the Experiment 4.4. Note that the "simple supernode" ablation model and the model used in (Pham et al., 2017) allow bi-directional message passings between original nodes and the supernode while the one used in (Li et al., 2017) only allows directional messages from the original nodes to the supernode.

For evaluation, we used a softmax cross entropy for classification tasks and a mean squared error for regression tasks. We use a fixed set of hyperparameters throughout the study.

All models were trained with Adam (Kingma & Ba, 2015). We report the results of the model snapshots of the epoch for which the best validation score was achieved. For implementation details including readouts and hyperparameters, please read the appendix.

### 4.3 TRAINING AND TEST LOSS REDUCTION

We explain the details of the experiment that produced the result presented in Fig. 1. For this experiment, we reported the average $\bar{r}$ of the loss reduction ratio $r = \frac{\mathscr{L} - \mathscr{L}^{(+)}}{\|L\|}$ for both training loss and test loss over 10 runs. $\mathscr{L}$ denotes the loss of the vanilla model, and $\mathscr{L}^{(+)}$ denote the loss of the GWM-installed model. $\bar{r}_{train}$ denotes a reduction ratio of the training loss, and $\bar{r}_{test}$ denotes a reduction ratio of the test loss.

Fig. 1 is the scatter plot of the $(\bar{r}_{train}, \bar{r}_{test})$, with the dotted slope representing $\bar{r}_{train} = \bar{r}_{test}$. For all GNNs and datasets, we set $L = 3$ and $D = 50$. As we can see in the plot, $\bar{r}_{train}$s were negative for the two GNNs in the HIV dataset (blue and brown circles). $\bar{r}_{train}$ for GGNN on QM9 (blue cross) was a very small negative value). For the other 13 (model-dataset) pairs, the attachment of GWM consistently reduces the training loss( i.e. improves the fit to the training graph datasets. ) Remarkably, 15 out of 16 pairs had positive $\bar{r}_{test}$s: the GWM improved generalization performances in most cases. It is worthy of note that $\bar{r}_{train}$ and $\bar{r}_{test}$ are positively correlated in this scatter plot. We were able to obtain similar results for all other choices of hyperparameters we tested. For the results with different hyperparameter values, see the appendix. This result implies that our GWM has the general effect of improving the generalization performance by augmenting the representation power of the main GNN.

### 4.4 EFFECT OF THE GWM ON THE REPRESENTATION POWER OF MODEL SPACE

In the second experiment, we studied the effect of GWM on the representation power of GNN models. To compare the GWM-augmented GNNs with their vanilla GNN counterparts on fair grounds, we used Bayesian optimization to optimize the number of layers and the dimension of feature vectors for each model-dataset pair we tested. Hyperparameters are optimized via the Optuna (Akiba et al., 2019) library.

We conducted a set of ablation studies to investigate the effect of (1) attention mechanism, (2) Gating mechanism, and (3) the Recurrent unit. We used two ablation models. **Simple supernode** model is a supernode without attention, gatings, and GRU functions. This ablation model can be considered a variant of (Pham et al., 2017)'s supernode model in which the supernode and the bulk networks communicate with each other in bidirectional manner. This model can be also considered as an extended version of supernode proposed in (Li et al., 2017).

**NoGate GWM** is a GWM without gatings, and it lacks GRU for the main GNN.

Table 1 summarizes the details of our ablation studies. For the detailed formulations of the two ablation models, please see the appendix.

Table 2 and Table 3 respectively present the MAEs on the regression tasks and the ROC-AUCs for the classification tasks, averaged over 10 random runs. In the tables, bold faces indicate the improvements from the vanilla GNN and asterisks indicate the best model among supernode models

| Dataset | GNN model | NFP | Weave | RGAT | GGNN | RSGCN | GIN | # Improved |
|---|---|---|---|---|---|---|---|---|
| LIPO | vanilla GNN | .677 | 1.19 | .753 | .582 | .801 | .844 | - |
| | +Simple Supernode | .693 | **1.01** | **.740** | .604 | **.775** | **.819** | 4/6 |
| | +NoGate GWM | **.675** | **.721** | **.688** | **.576** | **.787** | .847 | 5/6 |
| | +Proposed GWM | **.672*** | **.688*** | **.659*** | **.569*** | **.752*** | **.784*** | **6/6** |
| QM9 | vanilla GNN | 6.16 | 6.38 | 8.96 | 4.92 | 15.2 | 14.0 | - |
| | +Simple Supernode | 7.68 | **5.51** | 9.00 | 5.41 | **14.6** | **11.5*** | 3/6 |
| | +NoGate GWM | 6.84 | **5.40*** | 9.21 | 5.52 | **12.5** | 12.9 | 3/6 |
| | +Proposed GWM | **6.64*** | 5.90 | **8.39*** | **4.88*** | **11.9*** | 11.8 | **5/6** |

Table 2: MAEs on the LIPO dataset and QM9 dataset. Smaller values are better. Scores on QM9 are the average MAEs over 12 sub-tasks. The score of select models are presented in the appendix.

| Dataset | GNN model | NFP | Weave | RGAT | GGNN | RSGCN | GIN | # Improved |
|---|---|---|---|---|---|---|---|---|
| HIV | vanilla GNN | .724 | .670 | .707 | .746 | .746 | .729 | - |
| | +Simple supernode | .707 | **.676** | .704 | **.764*** | .728 | .729 | 2/6 |
| | +NoGate GWM | .714 | **.680** | .726 | .744 | .742 | **.739** | 3/6 |
| | +Proposed GWM | **.731*** | **.681*** | **.748*** | .762 | **.758*** | **.755*** | **6/6** |
| Tox21 | vanilla GNN | .763 | .710 | .764 | .757 | .760 | .740 | - |
| | +Simple supernode | **.770** | **.750** | **.787*** | **.790** | **.770*** | **.763** | 6/6 |
| | +NoGate GWM | **.775*** | **.764** | **.786** | **.792*** | .759 | **.766** | 5/6 |
| | +Proposed GWM | **.769** | **.767*** | **.787*** | **.785** | **.766** | **.768*** | **6/6** |

Table 3: ROC-AUCs on the HIV dataset and Tox21 dataset. Larger values are better. Scores on Tox21 are the average MAEs over 12 sub-tasks. Scores of select models are presented in the appendix.

for each (dataset, GNN) pair. Full tables with standard deviations are presented in the appendix. As we can see in the tables, the proposed GWM improves the generalization performances for 23 out of 24 (model-dataset) pairs. These results suggest that the proposed (full) GWM can improve GNNs' performances irrespective of the choice of GNN models and the dataset.

A few words of caution are in order here. Two ablation models did not improve the generalization of GNNs for the QM9 and the HIV datasets (see the column "# Improved"). This suggests an appropriate combination of attentions, gatings, and GRUs is essential in making the supernode effective for the analysis of molecular graph datasets.

We shall emphasize that the goal of this study is not to find the specific network architecture that achieves the states of the art performance for selected datasets[2]. Instead, the goal of our work is propose an attachable module that *improves* the representation power and a generalization performance *irrespective* of the choice of GNN architecture. As we can see in the presented result, the attachment of GWM improves the result in most cases; one of the results of our GWM attached model for Tox21 is actually SOTA (0.787, achieved by GWM attached RGAT).

## 5 CONCLUSION

For a generic DNN, numerous effective *installable modules* have been proposed for the improvement of the model (e.g. (Srivastava et al., 2014; Ioffe & Szegedy, 2015; Miyato et al., 2018; He et al., 2016)). The proposed GWM is the first of its kind to be installed to a generic GNN as an auxiliary module. Experimental results show that the GWM can generally improve the representation power as well as the generalization performance of a GNN, irrespective of the choice of GNN architecture and the molecular graph datasets. We would like to emphasize that the choice of the internal structure of GWM is not limited to the ones we described in this study, and that there are possibly numerous ways to construct a GWM-like module. For example, there is no provable justification for the use of a linear transformation in the transmissions or a bilinear form in the attention coefficients $\alpha$. Effective choices of supernode features are also open to further research. Our study can possibly open an entirely new avenue for the architectural study of GNNs.

---

[2]For example, Yang et al. (2019) and Li et al. (2017) achieve the SotA AUCs (0.776) of GNNs on the HIV dataset.

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

## A    OUR FORMULATION OF RGAT

Apart from the original RGAT (Busbridge et al., 2018), we have developed a similar GNN in a slightly different formulation. Followings are our RGAT formulation:

$$h_{\ell+1,i} = \tanh\left(W_l \textbf{concat}_{k=1}^{K} \tilde{h}_{\ell,i,k}\right), \tag{10}$$

$$\tilde{h}_{\ell,i,k} = F_{\ell,k} h_{\ell,i} + \sum_{j \in N_i} \alpha_{i,j,k} G_{\ell,k} h_{\ell,j}, \tag{11}$$

$$\alpha_{i,j,k} = \textbf{softmax}\left(a\left(h_{\ell,i}, h_{\ell,j}; A_{\ell,k,e_{i,j}}\right)\right). \tag{12}$$

$$a\left(h_{\ell,i}, h_{\ell,j}; A_{\ell,k,e_{i,j}}\right) = h_{\ell,i}^T A_{\ell,k,e_{i,j}} h_{\ell,j}. \tag{13}$$

$W, F, G, A$ are the coefficient matrix to be tuned. $\ell$ is the index of the layer up to $L$, $k$ is the index of the attention head up to $K$, $i, j$ are the index of the nodes up to $N$, $e_{i,j} = r$ is the index of the edge type up to $R$.

The main point is the edge type information in Eq.13. The edge type $e_{i,j} = r$ switches the weight matrix of the attention similarity function, $a$. This means that the associations between nodes should be computed dependent on the edge type. This is a natural assumption for chemical molecular graphs. Typically we have multiple bond types between nodes = atoms: single-bond, double-bound, triple-bond, and the aromatic ring. It is natural to assume that interactions between atoms are affected by the bond types among the atoms.

The main differences from the original RGAT lie in the Eq.11. The original RGAT assumes that the weight matrix $G$ is also dependent on the edge type ($G_{\ell,k,e_{i,j}}$) while we omit this dependency. Also, the original RGAT does not provide a self-link weight matrix $F$ while we do. We made these changes based on our preliminary experiments. We found our formulation is better than the original RGAT formulation in the MoleculeNet dataset, in terms of the training stability and the generalization performances.

Another difference is the choice of the attention function. In our formulation, the attention similarity measure $a(\cdot)$ is defined by the *general* attention in (Luong et al., 2015) while the original GAT (Veličković et al., 2018) and the RGAT (Busbridge et al., 2018) employed a simpler *concat* attention.

## B    OUR IMPLEMENTATION OF GIN

We implement the simplest GIN: 2-layer MLP with ReLU activation for each layer and a bias parameter $\epsilon$ fixed at 0. We regularize GIN with dropout (Srivastava et al., 2014), instead of batch-normalization (Ioffe & Szegedy, 2015). This is because the batch-normalization of the Chainer-Chemistry library did not correctly treat the padded node elements in the minibatches when we conducted the experiments.

## C    EXPERIMENTS DETAILS: GENERAL ISSUES

### C.1    GRAPH DATA REPRESENTATION

All datasets used in our experiments are taken from the MoleculeNet(Wu et al., 2018). Four used datasets are provided in the SMILES format. A SMILES format is a line notation for describing the

structure of chemical compounds. We decode a SMILES molecular data into a graph representation of the molecule. A node in the graph corresponds to an atom. Each atom node is associated with the symbolic label of the atom name ("H", "C", ...). An edge in the graph corresponds to a bond between atoms. Each bond edge is associated with the bond type information (single-, double-, ....).

Given the graph, we extract input feature vectors for node $x_i$ and that of supernode $x'$. $x_i$, the input feature vector for the node $i$ is a $D$-dimensional continuous vector, which is an embedded vector of the one-hot atom label vector with a trainable linear transformation. $X'$, The input feature vector for the supernode is a $D'$-dimensional continuous vector, which again is an embedded vector of some graph-global features with a trainable linear transformation. Choices for the graph-global features are detailed in the following section.

The edge information is converted in an adjacency matrix, $\mathscr{A}$.

## C.2 Explicit Features for Supernode

Since the supernode does not exist in the original graph $G$, we have no observable cues for the supernode. For simplicity, we propose to use an aggregation of node features, such as:

- Histograms of discrete labels attached to original nodes
- Averages, maximums, minimums, or medians of numerical attributes attached to original nodes
- Histograms of edge types if the graph is multiple relational graph.
- Number of nodes, graph diameters, modularity, and other simple statistics for graph structure.

We can augment the super feature vector $x'$ if some additional information about the graph is provided. Essentially, these simple aggregations of the feature vectors do not bring new information into the network. However we found that the graph-wise super feature input boosts the performance of the learned network model.

## C.3 Data Splits

In chemical datasets, a totally random shuffling of samples into train/val/test subsets is not always a valid way of data splitting. Therefore MoleculeNet provides several ways of data splitting. The "random" split is the random sample shuffling that are most familiar to the machine learning community. The "scaffold" split separate samples based on the molecular two-dimensional structure. Since the scaffold split separates structurally different molecules into different subsets, "it offers a greater challenge for learning algorithms than the random split" (Wu et al., 2018). Throughout the paper, we adopt the scaffold split to assess the full potential of the GWM-attaching GNNs.

The actual construction of the scaffold split train/validation/test subsets has a freedom of algorithm choices. We basically adopted the algorithm provided by the deepchem[3] library, which is the standard split algorithm for many papers. However, for the experiment of train/test loss comparison, we adopted the algorithm provided by the Chainer Chemistry library.

## C.4 Readout Layer

In many applications of GNNs users may expect a single fixed-length vector representing the characteristics of the graph $G$. So we add the 'readout' layer to aggregate the original node hidden states $\{H_\ell\}$ and the global node hidden states $\{g_\ell\}$.

A main issue in the readout unit is how to aggregate the original nodes, whose number varies for each graph. A simple way is to take an arithmetic average (sum) of the $h$s at the $L$-th layer, but we can also use a DNN to compute (non-linear) "average" of $h$s (Li et al., 2016; Gilmer et al., 2017). After the aggregation of the node hidden states, we simply concatenate it with $g$s and apply some transformations to achieve the readout vector, $r$:

$$r = \mathrm{DNN}_{r1}\left(\mathrm{concat}\left[\mathrm{DNN}_{r2}\left(H_L\right), g_L\right]\right). \tag{14}$$

---

[3]https://deepchem.io/

In the above equation, $\text{DNN}_{r1}$ is a multi-layer perceptron or a fully connected layer to mix the concatenated hidden vectors. We adopted a simple fully-connected layer for $\text{DNN}_{r1}$ in this paper. $\text{DNN}_{r2}$ is a specific readout unit accompanied with a original GNN to aggregate variable-length $H_L$.

## C.5 OPTIMIZER

All models were trained with Adam (Kingma & Ba, 2015), $\alpha = 0.001$, $\beta_1 = 0.9$, and $\beta_2 = 0.999$.

## C.6 ABLATION MODELS FORMULATION

Here we detail the formulation of the ablation models used in the main comparison experiments (Sec. 4.4).

As written in the main manuscript, we formulate the two ablation models (Table 1) as follows. A **simple supernode** model is an ablation model without attentions, gates, nor the GRUs, and it can be considered as a variant of (Li et al., 2017; Pham et al., 2017) with bidirectional communication between the supernode and the bulk, omitting all attentions and gates. First, there is no attention for the Transmitter. So the message from the main nodes to the supernode is just a sum of hidden vectors, $h_{\ell-1,:}$:

$$h_\ell^{\text{main}\to\text{super}} = \tanh\left(W_\ell \sum_i h_{\ell-1,i}\right) \in \mathbb{R}^{D'}. \tag{15}$$

Originally there is no messages from the supernode to the main GNN in (Li et al., 2017), but we allow such a simple message in this ablation model:

$$g_\ell^{\text{super}\to\text{main}} = \tanh\left(F_\ell g_{\ell-1}\right) \in \mathbb{R}^{D'}. \tag{16}$$

Messages are merged by simple linear combinations, instead of gates and GRUs, following (Li et al., 2017):

$$h_{\ell,i} = \boldsymbol{Z}_{\ell,1}\hat{h}_{\ell,i} + \boldsymbol{Z}_{\ell,2}g_\ell^{\text{super}\to\text{main}} \in \mathbb{R}^D, \tag{17}$$

$$g_\ell = \boldsymbol{Z}_{\ell,1}^{(S)} h_\ell^{\text{main}\to\text{super}} + \boldsymbol{Z}_{\ell,2}^{(S)}\hat{g}_\ell \in \mathbb{R}^{D'}. \tag{18}$$

We find it difficult to fully recover the supernode of (Gilmer et al., 2017) since their description on the supernode is quite limited. Thus, a **NoGate GWM** model, which surrogates (Gilmer et al., 2017), only capture the essence of their supernode: no gatings for merger, and GRU is not installed for the nodes of the main GNN. In this model, we use the same attention-based Transmitter unit as in Eqs.(2-6). We reduce the adaptive gatings in the Warp unit by a simple averaging, and omit the GRU for $h_{\ell,i}$.

$$h_{\ell,i}^0 = \boldsymbol{Z}_{\ell,1}\hat{h}_{\ell-1,i} + \boldsymbol{Z}_{\ell,2}g_\ell^{\text{super}\to\text{main}} \in \mathbb{R}^D, \tag{19}$$

$$g_\ell^0 = \boldsymbol{Z}_{\ell,1}^{(S)} h_\ell^{\text{main}\to\text{super}} + \boldsymbol{Z}_{\ell,2}^{(S)}\hat{g}_\ell \in \mathbb{R}^{D'}, \tag{20}$$

$$h_{\ell,i} = h_{\ell,i}^0 \in \mathbb{R}^D, \tag{21}$$

$$g_\ell = \textbf{GRU}\left(g_{\ell-1}, g_\ell^0\right) \in \mathbb{R}^{D'}. \tag{22}$$

## C.7 HYPERPARAMETER

We fix a part of hyperparameters throughout the experiments, which does not influence performances so much: the number of heads in all multi-head attention mechanisms to $K = 8$, and used $R = 4$ edge types for the multi-relational mechanism in all models. Also, at every layer, we set the dimension of the supernode feature to be the same as that of the features of the nodes in the main GNN.

In the next section, we list the other hyperparameters (the number of layers $L$, the dimension of feature vectors $D(= D')$) used in several experiments/figures, as well other experimental/implementation details.

## C.8 COMPUTATIONAL ENVIRONMENT

We use a single GPU (mainly nvidia Tesla V100) for an experimental run. A run roughly takes 1 hour to 1 day, depending on the hyperparameters and the GNN models.

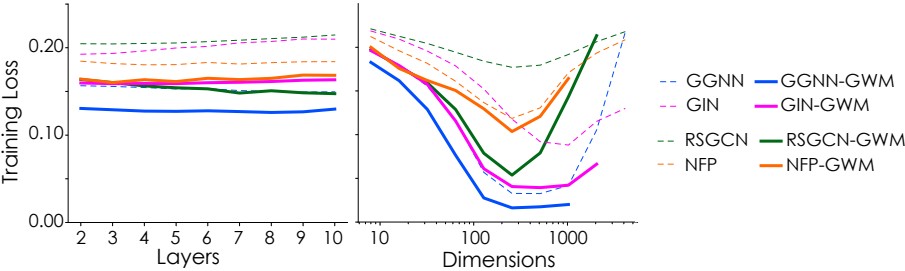

Figure 4: Training losses of various GNN models on a molecule graph dataset (Tox21). The horizontal axis denotes the number of GNN layers (the left panel) or the dimension of the node feature vectors (the right panel). Color denotes the GNN model. Thinner dashed lines are the losses of the vanilla GNNs, while thicker solid lines show the losses of the GNNs attached with the proposed Graph Warp Module ("GWM"). Scores are partially unavailable due to memory shortages.

## D EXPERIMENTS DETAILS: FOR EACH EXPERIMENT

In this section, we report details for each experiment, including the chosen hyperparameters and additional results.

### D.1 TRAINING LOSS BEHAVIORS OVER NUMBER OF LAYERS AND EMBEDDING DIMENSIONS

We used the Tox21 dataset to confirm behaviors of training losses, mentioned in the first paragraph of the introduction (Fig. 4).

To study the effect of the number of layers $L$ (the left panel), we fixed the dimension $D = 32$. To study the effect of the feature vector dimensions $D$ (the right panel), we fixed the number of layers $L = 4$.

All models are trained for 30 epochs.

Thinner dashed lines in Fig. 4 plot the training losses of original networks (w/o GWM). Thick solid lines in Fig. 4 plot the training losses of networks augmented with GWM. In general, GWM has an effect of decreasing the training loss for most choices of numbers of layers and dimensions of the node feature vectors, for all GNNs. In general, similar results are obtained from the other three datasets so we omit these figures.

### D.2 SECTION 4.3: TRAIN AND TEST LOSS REDUCTION

In the experiment in the Section 4.3 (Fig. 1), we align the hyperparameters including $L, D$ among a vanilla GNN and its GWM-installed counterpart, to compare the loss function values. We used four datasets. For each dataset, we fixed $L$ and $D$ for all GNN models to compare the loss reduction performances.

In Fig. 1, $L = 3$ and $D = 50$ for all GNNs and datasets. In preliminary experiments, we manually changed $Ls$ ($\in \{2, 3, 4\}$) and $Ds$ ($\in 32, 50, 100, 150$) in some extent, but found the overall tendency of the scatter plots does not dramatically change. This is partially understood from the Figure 1 in the main manuscript: the loss curves of the vanilla GNNs and their GWM-augmented counterparts evolve in roughly parallel. This implies the ratios of loss reductions are not so much dependent on the hyperparameter choices.

Here we show the results of other hyperparameter settings. All cases we observe the similar plot patterns.

Fig. 5 is the scatter plot of the $(\bar{r}_{train}, \bar{r}_{loss})$, $L = 3$ and $D = 32$. In this case, two HIV dataset plus one case for QM9 reported the increase of the training loss. For other 13 pairs, the GWM successfully reduce the training losses. It is remarkable that all 16 plots have positive test loss reduction rates: namely, the generalization performances are improved by the GWM in this choice of the hyperparameters.

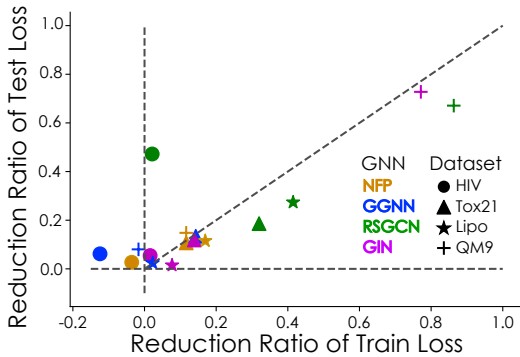

Figure 5: Train (horizontal) and test (vertical) loss reduction ratios on various pairs of GNN models and datasets. $L = 3$, $D = 32$. Each plot presents the rational train/test loss reductions induced by the GWM attachment for a specific pair of (dataset(symbol), GNN(color)).

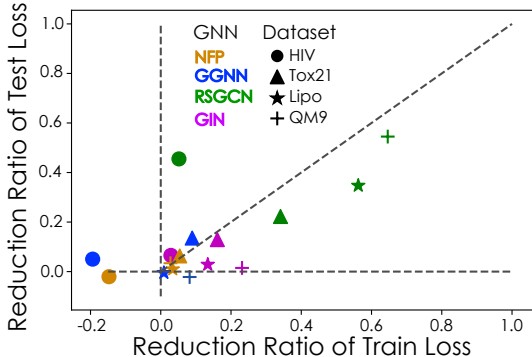

Figure 6: Train (horizontal) and test (vertical) loss reduction ratios on various pairs of GNN models and datasets. $L = 4$, $D = 100$. Each plot presents the rational train/test loss reductions induced by the GWM attachment for a specific pair of (dataset(symbol), GNN(color)).

Fig. 6 is the scatter plot of the $(\bar{r}_{train}, \bar{r}_{loss})$, $L = 4$ and $D = 100$. In this case, only two cases of the HIV datasets record the negative $\bar{r}_{train}$ values. For other 14 pairs, the GWM successfully reduce the training losses and 13 out of these 14 pairs have positive $\bar{r}_{test}$s.

All models were trained for 30 epochs.

## D.3 SECTION 4.4: THE FULL COMPARISON

For the experiments in the Section 4.3 (Table 2 and 3 in the main manuscript), we employed the Bayesian optimization to tune $L$ and $D$ for each combinations of a dataset and a GNN model. The Bayesian optimization (BO) trials were conducted by the Optuna library, with 200 sampling (searches) for each combination. Ranges of the BO search is: $2 \leq L \leq 8$, $4 \leq D \leq 512$.

Chosen $L$s and $D$s are presented in the Table 4.

Note that GWM-attached GNNs tend to perform better with deeper architecture (i.e larger $L$), when the vanilla GNNs without GWM do not change its overall performance with the number of layers (Table 4). The performance of the GWM-attached models often improve with the number of layers until the model depth reaches quite a large number ( 8) (e.g. [QM9, GGNN+Proposed GWM], [HIV, GIN+Proposed GWM]). Out of 24 pairs (=4 datasets × 6 GNN models), 11 pairs perform better with deeper architecture when deployed with GWM.

For the QM9 dataset, we trained the models for 50 epochs. For the HIV and the Tox21 dataset, we trained the models for 100 epochs. For the LIPO dataset, we trained the models for 200 epochs.

| Dataset | GNN model | NFP | WeaveNet | RGAT | GGNN | RSGCN | GIN |
|---------|-----------|-----|----------|------|------|-------|-----|
| LIPO | vanilla GNN | (4, 232) | (3, 50) | (4, 9) | (4, 32) | (5, 19) | (6, 19) |
|  | +Simple Supernode | (2, 71) | (3, 9) | (3, 8) | (4, 18) | (2, 27) | (3, 14) |
|  | +NoGate GWM | (2, 65) | (2, 14) | (3, 12) | (5, 30) | (2, 15) | (6, 9) |
|  | +Proposed GWM | (5, 231) | (4, 15) | (5, 19) | (6, 127) | (4, 22) | (2, 26) |
| QM9 | vanilla GNN | (5, 86) | (3, 22) | (4, 40) | (5, 71) | (4, 100) | (4, 250) |
|  | +Simple Supernode | (5, 44) | (3, 111) | (6, 34) | (5, 60) | (5, 69) | (3, 22) |
|  | +NoGate GWM | (4, 48) | (5, 250) | (4, 31) | (6, 50) | (4, 48) | (4, 34) |
|  | +Proposed GWM | (5, 72) | (3, 104) | (4, 156) | (8, 50) | (4, 36) | (4, 23) |
| HIV | vanilla GNN | (6, 213) | (2, 65) | (3, 28) | (6, 29) | (4, 57) | (2, 72) |
|  | +Simple supernode | (3, 30) | (3, 39) | (2, 9) | (4, 54) | (4, 255) | (5, 93) |
|  | +NoGate GWM | (4, 46) | (3, 20) | (2, 12) | (2, 51) | (3, 27) | (3, 60) |
|  | +Proposed GWM | (3, 200) | (3, 92) | (3, 23) | (8, 135) | (3, 106) | (8, 38) |
| Tox21 | vanilla GNN | (3, 204) | (5, 90) | (3, 19) | (6, 36) | (5, 70) | (5, 103) |
|  | +Simple supernode | (5, 129) | (6, 31) | (2, 36) | (6, 136) | (5, 119) | (5, 157) |
|  | +NoGate GWM | (2, 123) | (2, 157) | (3, 43) | (5, 79) | (4, 31) | (6, 117) |
|  | +Proposed GWM | (3, 106) | (4, 19) | (3, 37) | (7, 48) | (8, 32) | (6, 102) |

Table 4: Hyperparameters $L$ and $D$ for the experiment in Section 4.4. The format of the table cells is: $(L, D)$.

Tables 5 and 6 are the full lists of the main comparison experiments in the main manuscript, with the standard deviation values in parentheses.

| Dataset | GNN model | NFP | WeaveNet | RGAT | GGNN | RSGCN | GIN |
|---|---|---|---|---|---|---|---|
| LIPO | vanilla GNN | .677 (.040) | 1.19 (.327) | .753 (.045) | .582 (.022) | .801 (.014) | .844 (.026) |
|  | +Simple Supernode | .693 (.027) | 1.01 (.208) | .740 (.032) | .604 (.027) | .775 (.011) | .819 (.023) |
|  | +NoGate GWM | .675 (.017) | .721 (.225) | .688 (.013) | .576 (.022) | .787 (.048) | .847 (.048) |
|  | +Proposed GWM | .672* (.040) | .688* (.105) | .659* (.016) | .569* (.022) | .752* (.014) | .784* (.012) |
| QM9 | vanilla GNN | 6.16 (.231) | 6.38 (.289) | 8.96 (.192) | 4.92 (.145) | 15.2 (1.05) | 14.0 (2.47) |
|  | +Simple Supernode | 7.68 (.316) | 5.51 (.248) | 9.00 (.399) | 5.41 (.110) | 14.6 (1.71) | 11.5* (.755) |
|  | +NoGate GWM | 6.84 (.348) | 5.40* (.238) | 9.21 (.368) | 5.52 (.344) | 12.5 (.564) | 12.9 (1.32) |
|  | +Proposed GWM | 6.64* (.360) | 5.90 (.248) | 8.39* (.219) | 4.88* (.231) | 11.9* (1.48) | 11.8 (1.21) |

Table 5: MAEs on the LIPO dataset and QM9 dataset. The number of layers and the dimension of the feature vectors are defined via Bayesian Optimization for each method and each dataset. Averages (standard deviations) over 10 random runs. Smaller values are better.

| Dataset | GNN model | NFP | WeaveNet | RGAT | GGNN | RSGCN | GIN |
|---|---|---|---|---|---|---|---|
| HIV | vanilla GNN | .724 (.017) | .670 (.020) | .707 (.039) | .746 (.018) | .746 (.011) | .729 (.020) |
|  | +Simple supernode | .707 (.023) | .676 (.031) | .704 (.019) | .764* (.014) | .728 (.009) | .729 (.025) |
|  | +NoGate GWM | .714 (.018) | .680 (.043) | .726 (.024) | .744 (.006) | .742 (.019) | .739 (.012) |
|  | +Proposed GWM | .731* (.020) | .681* (.010) | .748* (.019) | .762 (.016) | .758* (.022) | .755* (.009) |
| Tox21 | vanilla GNN | .763 (.004) | .710 (.029) | .744 (.008) | .764 (.009) | .760 (.005) | .740 (.007) |
|  | +Simple supernode | .770 (.008) | .750 (.020) | .787* (.005) | .790 (.006) | .770* (.006) | .763 (.006) |
|  | +NoGate GWM | .775* (.007) | .764 (.011) | .786 (.009) | .792* (.008) | .759 (.007) | .766 (.008) |
|  | +Proposed GWM | .769 (.007) | .767* (.021) | .787* (.009) | .785 (.005) | .769 (.006) | .768* (.007) |

Table 6: ROC-AUCs on the HIV dataset and Tox21 dataset. The number of layers and the dimension of feature vectors are defined via Bayesian Optimization for each method and each dataset. Averages (standard deviations) over 10 random runs. Larger values are better.

Finally we present the detailed reports on the sub-tasks of QM9 and Tox21. These datasets consists of 12 sub-tasks and so far we reported the sub-task-averaged scores. Below we report the scores changes brought by GWM-attached GNNs in Table 7 (QM9) and Table 8 (Tox21). For QM9, we report the relative reductions of MAE in percentage(%). For Tox21, we report the improvements of binary classification accuracy in percentage (%). Numbers of sub-tasks improved by GWM (positive value slots) are in good accordance with the score gains in Table 2 and Table 3.

| GNN/Tasks | mu | alpha | HOMO | LUMO | gap | r2 | zpve | cv | u0 | u298 | h298 | g298 |
|-----------|-----|-------|------|------|-----|-----|------|-----|------|------|------|------|
| GGNN | -0.6 | -0.9 | 1.0 | 1.6 | 4.6 | 2.9 | 19 | 0.3 | -3.9 | -4.8 | -8.3 | -4.2 |
| WeaveNet | -2.2 | -0.3 | 27 | -35 | 52 | 6.7 | 107 | -11 | 84 | 91 | 86 | 87 |

Table 7: Relative reduction of MAE (%) GWM, 12 tasks in QM9. Larger values are better.

| GNN/Tasks | Task1 | Task2 | Task3 | Task4 | Task5 | Task6 |
|-----------|-------|-------|-------|-------|-------|-------|
| RGAT | 0.9 | 0.2 | 0.7 | 0.5 | 1.2 | 0.2 |
| GGNN | 0.0 | 3.7 | 2.1 | 1.1 | 1.5 | 16 |
| GNN/Tasks | Task7 | Task8 | Task9 | Task10 | Task11 | Task12 |
| RGAT | -0.5 | 1.2 | -0.0 | 0.2 | 1.9 | 0.4 |
| GGNN | 5.9 | 2.8 | 0.7 | 1.0 | 2.4 | 1.5 |

Table 8: Improvements of binary classification accuracy (%) by GWM, 12 tasks in Tox21. Larger values are better.

