# OpenReview forum: "Graph Warp Module: an Auxiliary Module for Boosting the Power of Graph Neural Networks in Molecular Graph Analysis"
_ICLR.cc/2020/Conference — Reject_

### Official Review · AnonReviewer2 · 2019-10-18
**Official Blind Review #2**

**Rating:** 6

**Review:**

= Summary
A Graph Neural Network extension integrating a global supernode explicitly into the message passing process. Concretely, message passing along the graph edges is alternated with message passing to/from a fresh super-node. Experiments show that this improves results of a number of common GNN architectures on four datasets.

= Strong/Weak Points
+ Simple but useful extension of the existing super-node idea
+ Experiments on a number of datasets and baseline GNN architectures, providing ample experimental evidence of the usefulness of the method.
- Writing is overcomplicated and uses a lot of jargon ("transmitter unit", "warp gate", "intermodule hyperspace"). I found the text entirely impenetrable and instead simply focused on Fig. 3 + the actual equations.

= Recommendation
This is a nice contribution of minor novelty, with empirical evidence of its usefulness. I believe the paper should be accepted to a large conference such as ICLR.

= Minor Comments
- Fig. 3: Inconsistent "intra-module" (top) vs. "intra module" (bottom)
- Concurrent work in https://openreview.net/forum?id=B1lnbRNtwr discusses a "sandwich" model which alternates graph message passing with (essentially) a Transformer layer applied to all nodes. This idea seems related (in that it alternates local and global information exchange).

**Experience Assessment:**

I have published in this field for several years.

**Review Assessment: Checking Correctness Of Derivations And Theory:**

N/A

**Review Assessment: Checking Correctness Of Experiments:**

I assessed the sensibility of the experiments.

**Review Assessment: Thoroughness In Paper Reading:**

N/A

---

> ### Author Response · Authors · 2019-11-11
> **Reply to the official blind review #2**
>
> Thank you very much for positive comments and reviews.
> As for the word-choices, we might change the naming of the submodules and expression in order to make the writing easier to read.
>
> Also, thank you very much for the information about  https://openreview.net/forum?id=B1lnbRNtwr.
> We indeed believe that there is a connection to our work.
> One notable difference is that our supernode submodule has a network of its own with its own memory,  and that we are giving more freedom to how the global information with the local information is mixed.
> We added this discussion in the updated manuscript (Sec. 2.2).

---

### Official Review · AnonReviewer3 · 2019-10-26
**Official Blind Review #3**

**Rating:** 3

**Review:**

The paper proposes an auxiliary module for GNNs to boost the representation power. The new module consists of virtual supernode, attention unit, and gating unit, each of which is demonstrated useful in the experiments. The module can be applied to various types of GNNs.

This work can be seen as an improvement to previous virtual supernode based methods. Adding the attention units and gating units is rational, and the effectiveness is also proved in the ablation studies. However, the claimed contribution of improving the representation power may mainly come from the idea of supernodes (instead of the attention and gating). This largely reduces the novelty of this paper and make it incremental, because using virtual supernodes is not this paper’s original idea.

The paper is generally well written. However, the comparison with previous supernode based models is not described clearly enough. The authors listed the difference from (Glimer et al. 2017) and (Li et al. 2017) in Table 1, but ignored (Pham et al. 2017) and (Battaglia et al. 2018), which were also cited in the related work. Moreover, (Li et al. 2017)’s method is actually different from the simple supernode baseline, in that it is not a bidirectional message passing between supernode and the main network. Table 1 does not contain this property.


**Experience Assessment:**

I have published one or two papers in this area.

**Review Assessment: Checking Correctness Of Derivations And Theory:**

I did not assess the derivations or theory.

**Review Assessment: Checking Correctness Of Experiments:**

I assessed the sensibility of the experiments.

**Review Assessment: Thoroughness In Paper Reading:**

I read the paper at least twice and used my best judgement in assessing the paper.

---

> ### Author Response · Authors · 2019-11-11
> **Reply to the official review #3**
>
> Thank you very much for the comments!
> While we believe that supernode is an essential component of our invention, we also believe that supernode alone is not sufficient.
> For this claim,  please notice that our method performs much better than the ablation model with no attention/gate mechanism (simple supernode), which is, in essence, the same as an architecture proposed by (Pham et al  2017).
> In order to make this point more clear, we added explanations and discussions about the relationship of (Pham et al. 2017) and our ablated model (simple supernode) in the updated manuscript (Table1, Sec. 4.4, and the appendix C.6. )
>
> Also, as far as we understand,  (Battaglia et al 2018) is a survey paper, and that they do not make a proposal for a new algorithm in their work.
> Finally, thank you very much for pointing out our imprecise description of (Li et al 2017).
> We fixed the relevant phrases in Table 1 and Sec 4.4.

---

### Official Review · AnonReviewer1 · 2019-10-27
**Official Blind Review #1**

**Rating:** 6

**Review:**

Graph Neural Networks is a popular architecture for the analysis of chemical molecules. The authors propose an auxiliary module that can be attached to a GNN that can boost the representation power of GNNs. The auxiliary module has three building blocks: 1. a supernode, 2. a transmitter unit and 3. a warp gate unit. The authors show through carefully designed experiments that these additions can be attached to any type of GNN and that they are successful in reducing both the training error and test error. A variety of graph regression and graph classification tasks are chosen to show the efficacy of the method.

The paper is well written and easy to follow. The modification suggested by the authors is novel and useful. Experiments are well designed. An aspect that is not clear from the paper is the ability of these models to overfit the data as you increase the representation power of the network. While the authors claim that to be one of the shortcomings of existing GNNs, it is not clear whether the proposed method solves that problem. For example, in figure 4. the training loss hardly decreases as the number of layers are increased. It would be good if the authors can share any insights on this point.

Overall, I think this is a good paper that the community will benefit from.

**Experience Assessment:**

I have published one or two papers in this area.

**Review Assessment: Checking Correctness Of Derivations And Theory:**

I assessed the sensibility of the derivations and theory.

**Review Assessment: Checking Correctness Of Experiments:**

I carefully checked the experiments.

**Review Assessment: Thoroughness In Paper Reading:**

N/A

---

> ### Author Response · Authors · 2019-11-11
> **Reply to the official blind review #1**
>
> Thank you for your positive reviews and comments.
> As for your concern about the “ability of our GWM to overfit”, we do not intend to make as strong a statement as to say that GWM can help assure that one can indefinitely increase the model’s representation power by adding more and more layers.
> However, as can be seen in Table 4, unlike the vanilla models,  the performance of the GWM-attached models improve with the number of layers until the model depth reaches quite a large number (~8) (e.g. [QM9, GGNN+Proposed GWM], [HIV, GIN+Proposed GWM]).
> Out of 24 pairs of (dataset, GNN), 11 pairs favor much deeper architecture when deployed with GWM.
> We added this discussion in the updated manuscript (Appendix D.4)

---

> > ### Comment · AnonReviewer1 · 2019-11-11
> > **Thanks**
> >
> > Thank you for the clarification and for updating the paper.

---

### Decision · Program_Chairs · 2019-12-19

**Decision:**

Reject

**Comment:**

This paper presents an auxiliary module to boost the representation power of GNNs. The new module consists of virtual supernode, attention unit, and warp gate unit. The usefulness of each component is shown in well-organized experiments.
This is the very borderline paper with split scores. While all reviewers basically agree that the empirical findings in the paper are interesting and could be valuable to the community, one reviewer raised concern regarding the incremental novelty of the method, which is also understood by other reviewers. The impression was not changed through authors’ response and reviewer discussion, and there is no strong opinion to champion the paper. Therefore, I’d like to recommend rejection this time.